# Unraveling Latent Aspects of Urban Expansion: Desertification Risk Reveals More

**DOI:** 10.3390/ijerph17114001

**Published:** 2020-06-04

**Authors:** Gianluca Egidi, Ilaria Zambon, Ilaria Tombolin, Luca Salvati, Sirio Cividino, Samaneh Seifollahi-Aghmiuni, Zahra Kalantari

**Affiliations:** 1Department of Agricultural and Forestry Sciences (DAFNE), University of Tuscia, 01100 Viterbo, Italy; egidi.gianluca@unitus.it (G.E.); ilaria.zambon@unitus.it (I.Z.); 2Department of Architecture and Project, Sapienza University, 00100 Rome, Italy; ilaria.tombolini@uniroma1.it; 3Council for Agricultural Research and Economics (CREA), 52100 Arezzo, Italy; 4Department of Agriculture, University of Udine, 33100 Udine, Italy; sirio.cividino@uniud.it; 5Department of Physical Geography and Bolin Centre for Climate Research, Stockholm University, 104 65 Stockholm, Sweden; zahra.kalantari@natgeo.su.se

**Keywords:** suburbs, land degradation, indicators, land use planning, Mediterranean Europe

## Abstract

Urban expansion results in socioeconomic transformations with relevant impacts for peri-urban soils, leading to environmental concerns about land degradation and increased desertification risk in ecologically fragile districts. Spatial planning can help achieve sustainable land-use patterns and identify alternative locations for settlements and infrastructure. However, it is sometimes unable to comprehend and manage complex processes in metropolitan developments, fueling unregulated and mainly dispersed urban expansion on land with less stringent building constraints. Using the Mediterranean cities of Barcelona and Rome as examples of intense urbanization and ecological fragility, the present study investigated whether land use planning in these cities is (directly or indirectly) oriented towards conservation of soil quality and mitigation of desertification risk. Empirical results obtained using composite, geo-referenced indices of soil quality (SQI) and sensitivity to land desertification (SDI), integrated with high-resolution land zoning maps, indicated that land devoted to natural and semi-natural uses has lower soil quality in both contexts. The highest values of SDI, indicating high sensitivity to desertification, were observed in fringe areas with medium-high population density and settlement expansion. These findings reveal processes of land take involving buildable soils, sometimes of high quality, and surrounding landscapes in both cities. Overall, the results in this study can help inform land use planers and policymakers for conservation of high-quality soils, especially under weak (or partial) regulatory constraints.

## 1. Introduction

Urbanization plays a key role in land take and soil consumption worldwide [1,2]. Rural areas are the socioeconomic context experiencing the most intense and widely investigated environmental impacts of urbanization [3,4,5]. More specifically, rural land is increasingly being converted to residential, commercial, and industrial settlements, producing socially polarized and economically unspecialized spaces, based on empirical evidence collected in both advanced economies and emerging countries [6,7,8]. Recent processes of urban expansion increasingly involve productive and high-quality rural contexts in advanced economies, seriously threatening natural landscapes [9].

While strongly interrelated, basic notions such as soil quality, land degradation, and desertification risk can be defined and characterized separately [10]. Land is a basic economic capital [11], as high-quality soils have ensured historical maintenance of feasible agriculture [12]. The concept of ‘soil quality’ is therefore complex, and several definitions have recently proposed associating it with the operational concepts of ‘suitability for use’ and ‘functionality’ [13]. The latter links soil quality to a more general ability of a given soil to perform the functions necessary for its intended use. A more comprehensive definition of soil quality is as the ability of a specific type of soil to perform functions supporting the productivity of plants and animals, maintaining or increasing air and water quality within natural or semi-natural ecosystems [14]. This definition emphasizes the value of soil in supporting ecosystem functionality and implies an explicit judgment on soil conditions meeting the principles of (environmental and socioeconomic) sustainability [15]. Its link with sustainability configures soil quality not only as an abstract concept, but also as a management objective to be pursued [16,17], integrating joint urbanization dynamics and ecological aspects into landscape governance, and thus linking soil quality with the more general notion of land quality [18].

Land supports ecosystem functions, thanks to the intrinsic ability to recover from biophysical and anthropogenic shocks [19]. The intrinsic value of the biophysical environment depends on a set of processes and evaluations, which are guided by humans and impact regional (socioeconomic) structures and soils [9,20,21]. With large-scale conversion of natural and agricultural areas to urban settlements, soil capacity to supply essential ecosystem services is decreased [21,22,23], through degradation of physical, chemical, and biological properties [24]. If human pressure exceeds certain limits, soil may no longer be able to perform some key functions and may become sensitive to degradation or, worst, desertification [25,26]. Land degradation is perceived as a key environmental issue for the 21st century owing to its consequences, such as effects on agronomic productivity and environmental systems [27,28,29]. In remediation approaches and policy-oriented literature, land degradation is interpreted as the joint outcome of physical and human interactions that progressively reduce the productive capacity of ecosystem services deriving from land [30]. Loss of ecosystem services dependent on land, at a higher scale, would have a negative effect on achievement of sustainable development goals defined in the 2030 Agenda [31].

The European Commission (EC) has proposed measures to tackle environmental and social issues linked to urbanization, soil sealing, and land degradation [32,33,34]. For instance, in its Thematic Strategy for Soil Protection, the EC underlines the need to develop best practices aimed at mitigating the negative effects of sealing on soil functions [34]. In 2012, the EC published a report on the most effective mechanisms for limiting, mitigating, or compensating for soil sealing [32]. In 2014, it published a study assessing the feasibility of setting up a framework for measuring progress towards more sustainable use of land [33]. It has been suggested that urbanization is responsible for the ‘consumption’ of fertile soils that are vital for agriculture and food production [35].

Land use planning is an appropriate tool for achieving more sustainable use of land and influencing urban expansion over time. For instance, using nature-based solutions (NBSs) in land use planning provides cost-effective long-term solutions for land degradation [36]. Exploring the potential of NBSs and employing them for land-related risk mitigation require in turn improved land use planning and management strategies [37]. In addition, flood risk in urban areas might increase under the impact of land use changes. Conversion of natural areas to impermeable surfaces for urban expansion results in lower infiltration rates and increased surface runoff, which in turn increase the flooding risks in urban regions and threaten urban expansion [38]. Considering the mutual interactions between land use planning and urban expansion, tackling land degradation is still a challenge for land use planners, land system scientists, and policymakers [39,40].

Land use planning should take into account the quality and characteristics of different land areas and soil functions, and balance them against competing objectives and private interests, e.g., those of urban developers. Various tools, such as decision support systems, geographic information systems (GISs), and socioeconomic/environmental indicators (or composite indices), have been designed to help authorities and land use planners understand complex urban systems, and plan for more sustainable and resilient future cities [37,41,42]. However, they are often not used in practice in land use planning, despite being available and potentially very useful [42]. In addition to identifying existing problems and patterns, application of these tools can enhance understanding among local and regional planners about the potential impacts of future urban expansion and planning decisions on the environment and urban containment [43].

Since the 1990s, strategic spatial planning (SSP) [44] has been increasingly undertaken at both urban and regional levels. In the past two decades, the common objective of SSP has been identification of a coherent spatial development strategy to frame medium- and long-term development of metropolitan regions [45,46]. This requires adoption of an integrated spatial logic regarding land use, preservation of natural resources, and major infrastructure development, e.g., housing and transportation [47,48]. SSP tools are therefore used as interpretative keys to investigate the progressive development of conservation measures in favor of land quality. SSP is mainly intended as an integrated and more sustainable development approach, involving various actors and being essentially multidimensional. When incorporated into the socio-political and institutional complexity in the real world, SSP is also influenced by power configurations and governance agreements [30]. Furthermore, SSP processes are often non-binding, and are thus less tailored to legally binding land use planning and policy instruments. They offer more advantages in a systematic review and provide generalizations that can help push scientific frontiers and policymaking agendas.

Integration of environmental protection measures with SSP has recently been attempted in some parts of Europe, especially in Northern, Western and Central European countries [3,9]. In other European regions, e.g., in the Mediterranean and Eastern areas, SSP is applied only occasionally, most likely due to less general societal awareness of land use planning benefits. In Southern Europe in particular, informal settlements and deregulated urban expansion from the 1950’s onwards, and more drastically from the 1960s to 1980s, were representative of particular socioeconomic systems, converging just partly toward a unified European system of land use planning that has been applied in recent years. Although prerequisites and background contexts differ between European cities, such patterns of urban expansion seem to be common to most of the countries in the European Union (EU), e.g., in some Eastern European countries and new member states, and in non-EU countries (Turkey, Israel, Southern Mediterranean countries, and some emerging Middle East countries).

Against this background, the aim of the present study was to investigate whether land use planning in two Mediterranean cities (Rome and Barcelona) is (directly or indirectly) conceived from a strategic perspective oriented towards conservation and enhancement of the environmental quality of soil resources. These cities were selected because they are surrounded by traditional rural landscapes (the Mediterranean agro-forest mosaic typical of lowland/coastal districts and mixing extensive tree crops (olives and vineyards), arable and garden crops, and relict woodland), experiencing increasing ecological fragility and land sensitivity to degradation under climate change and human pressures [49]. A commonly used methodology for assessing land quality, sustainable land management, and urban containment was used for both case cities. Official land zoning data sources for the two cities differ moderately at the local scale, because of differences in the size of administrative units. However, based on common planning tools, a standardization of land zoning classes was adopted for comparing Rome and Barcelona as two relevant examples of urban expansion in a region sensitive to land degradation (Mediterranean Europe).

## 2. Materials and Methods

### 2.1. Study Area

The cities of Rome and Barcelona are geographically similar (e.g., located at similar latitude and sharing past urban expansion processes), but differ in size and recent urban management. The city of Rome, partitioned into 19 urban districts, covers nearly 1285 km^2^ and its municipal territory incorporates rather fertile areas cultivated for centuries and, in some cases, left abandoned (or uncultivated) in more recent decades. Compared with other Mediterranean cities (e.g., Athens, Naples, Madrid, Salonika), the population density in Rome is relatively low (2233 inhabitants/km^2^) [48], because of the presence of green land scattered throughout the city. The municipal territory is heterogeneous, with mixed impervious and semi-natural land contrasting with the compactness of the historical center, where the most important functions are concentrated. However, with residential mobility and suburbanization, Rome’s morphology has been progressively transformed toward more polycentric and spatially balanced settlements.

Urbanization in Rome has frequently involved out-of-plan land, with partial regulatory constraints or with mixed/ambiguous destination. Land consumption in Rome was high during both the ‘compact expansion’ driven by population increase (1950–1990), and the more recent expansion (1990–2020) characterized by stable population and dispersed urban expansion. Rome’s expansion began with informal settlements in the 1950s and 1960s, indicating how post-war urbanization has manifested in a (partly) deregulated urban context, lacking an effective (and truly participatory) land use planning framework [48]. Despite more recent urban expansion [50], Rome is a city with high levels of urban congestion, population concentration, and economic polarization. The resident population in the city core has decreased only recently, as observed earlier in other Mediterranean cities [4]. Rome’s master plan for the period 1993–2008 identified tourism and culture as two main sectors promoting urban expansion. In 2008, after more than 40 years, Rome approved a new strategic master plan incorporating rules and guidelines to orient metropolitan development towards a more coherent urban design [18], devoting attention to issues of decentralization and polycentric urban functions, provision of adequate services in suburban areas, environmental protection, and cultural and historical heritage [50].

Barcelona, where the city center is partitioned into 10 urban districts, covers an area of 101.4 km^2^ and has a population density of 15,984 inhabitants/km^2^ [5]. Topography has played a major role in the evolution of the urban form of Barcelona, being dominated by two mountain ranges (the littoral range, with elevation above 700 m, and the pre-littoral range, with elevation above 1700 m) and two flat areas along the valleys of the Llobregat and Besos rivers [51,52,53]. The actual urban form is shaped by the progressive saturation of the inner core, which limits future expansion of Barcelona city and its conurbations [2]. Soil occupation, loss of agricultural and forest land, decreased settlement density, and a large amount of bare land awaiting further development are all important signals of landscape transformation [7,54]. Land-use changes in the area testify to outward expansion of the consolidated area, thanks to high levels of car ownership, relocation of industrial and retail activities to fringe land, development of transport infrastructure, and conversion of second homes into primary residences [3].

Based on a dedicated spatial analysis carried out through elaboration of maps provided by the European Environment Agency (EEA) (Source: Land imperviousness map, GMES Land Copernicus Programme), the city of Barcelona appears to be strongly urbanized, while the city of Rome is much more expansive, with natural areas surrounding the urban heart of the Italian capital (Figure 1). While the two cities differ in terms of size, population density, urban expansion capacity, and local socio-economic contexts, their experiences of past urban expansion processes are comparable. They are also both competent with regard to land use planning and land zoning considerations, which makes them substantially comparable regarding their administrative prerogatives. Overall, their comparison in this study provides a refined overview of the relationship between land use planning, land zoning systems, soil quality, and desertification risk in different morphological and functional contexts across Southern Europe.

### 2.2. Data, Variables and Indices

#### 2.2.1. Land Use Planning and Land Zoning

For the present analysis, land use classes were extrapolated from digital maps (shapefiles) illustrating the most recent strategic master plan of the two case cities. For Rome, these were extracted from the ‘Systems and Components’ of the town master plan approved by the City Council (Resolution no. 18, dated 12 February 2008), which governs the physical and functional transformation of the city [36]. It pursues the objectives of territorial redevelopment and enhancement, according to the principles of environmental sustainability and in compliance with the criteria of effectiveness, publicity, and simplification of administrative action, within the framework of current legislation. In the case of Barcelona, the Urban Map of Catalonia (MUC) [5] is used at the municipal scale as a strategic tool for implementation of planning policies [55,56]. It includes all general planning regulations in force on 1 January 2018 in the territory of Catalonia and those in force on 1 July 2017 in the metropolitan area of Barcelona [57,58]. The MUC is a summary map that allows investigation of the basic attributes of land use planning in the region, contributing to solve the inherent differences in codification, language, and representation characterizing Barcelona’s regional plans [59].

#### 2.2.2. Soil Quality

The environmentally sensitive area (ESA) framework [9,15,17] was used here to perform a comparative and comprehensive assessment of soil quality at the local scale in Rome and Barcelona, following the method applied by [60]. To assess soil resources, the soil quality index (SQI) proposed by EEA and based on [61] was calculated using information contained in the European Soil Database produced by the Joint Research Centre [62]. SQI is a composite index based on four variables: parent material, soil depth, soil texture, and slope angle. A set of sensitivity scores derived from statistical analysis and fieldwork performed by authors cited in [61] was assigned to each variable analyzed. SQI was then estimated as the geometric mean of the different scores attributed to the four variables, ranging from 1 (indicating the lowest contribution to land degradation sensitivity, and thus the highest soil quality) to 2 (indicating the highest contribution to land degradation sensitivity, and thus the lowest soil quality). SQI data are available in raster format and disseminated at 1 km^2^ spatial resolution [40]. Despite its acknowledged importance as a tool for assessing soil quality, the SQI approach has certain shortcomings because of the restricted number of variables considered [63,64]. For this reason, in the present study, an additional indicator of land quality and degradation was included (see Section 2.2.3), with the aim of obtaining a more general and ‘holistic’ assessment of desertification risk. This indicator is based on soil functions such as fertility, given the contribution of physical and chemical attributes and of external environmental factors to overall soil health.

#### 2.2.3. Land Sensitivity to Desertification

Land quality is a multi-dimensional component associated with land degradation processes. It is intimately related with the socio-economic context at the local scale and represents the ability of a particular kind of soil to sustain agricultural production and/or natural vegetation [65]. A sensitivity to desertification index (SDI) for use in the ESA framework to assess the level of land quality and susceptibility to degradation has been developed and validated in the field [66]. The composite SDI indicator provides a more comprehensive assessment of land quality, considering together the dimensions of climate, soil, and land use. A full description of the methodology can be found in [67]. The value of SDI ranges between 1 (highest land quality, lowest sensitivity to degradation based on the local environmental context) and 2 (lowest land quality, highest sensitivity to degradation). EEA has prepared a raster map (with resolution of 1 km^2^ grid) of SDI providing homogeneous coverage of the entire Mediterranean Europe region, based on computation of nine biophysical layers: four variables assessing soil quality (parental material, soil depth, texture, slope), climate quality based on the aridity index (ratio of annual precipitation to annual reference evapotranspiration rate) and four variables assessing vegetation quality (protection from soil erosion, resistance to drought, plant cover, resistance to fire) [67]. Input layers for the raster map were derived from official data sources referring to the late 1990s and covering the Mediterranean Europe regions at fine spatial resolution [67]. Values of each layer are ranked on a scale of 1–2 and the SDI is calculated as the geometric mean of the score of all input layers.

### 2.3. Analysis Framework Development

A nomenclature system was used here to standardize land zoning classes extrapolated from shapefile maps illustrating the strategic master plans for Rome and Barcelona elaborated by the respective town councils (Table 1 and Figure 2). In line with indications provided in earlier studies [50], the five classes identified were:Class 1: Conservation-protection land regulated by restrictions or with constrained urban development (red in Figure 2).Class 2: Consolidated urban fabric (grey in Figure 2) concentrating on the most compact and dense urban area in the two case cities. Consolidated urban areas, including the historical city, as well as service and infrastructural systems, coincided with the most dense settlements, with non-urbanized land mainly used for commercial activities.Class 3: Restructuring (urban and non-urban) areas, which are expected to be restored and redeveloped, including urban infill (yellow in Figure 2).Class 4: Transforming (non)urban areas, which will undergo residential development (orange in Figure 2).Class 5: The environmental system including green spaces, protected natural areas, water and coastal environments (green in Figure 2). This class includes rural (agricultural) areas (e.g., the ‘Agro Romano’, the traditional countryside around the city of Rome, and the ‘Collserola’ park and a small part of the agricultural land in the Llobregat Agrarian park in Barcelona).

#### Data Analysis

To compare the two study contexts, separate raster files containing geo-spatial information on SQI and SDI were re-classified using ArcGIS 10.5.1 [63], defining homogeneous classes for the two indicators. Since the aim of this study was to investigate whether land use planning considers the characteristics of different land areas and soil functions, the raster maps of SQI and SDI were then overlaid separately with the planned land use maps (shapefiles) for Rome and Barcelona. SQI values were categorized into three levels: high, intermediate, and low soil quality. SDI was organized into four levels. As mentioned, SDI values range between 1 (highest land quality, lowest sensitivity to degradation based on the local environmental context) and 2 (lowest land quality, highest sensitivity to degradation). In both case cities, SDI had an average value of between 1.1 and 1.3, reflecting rather high land quality. Four levels of SDI were then structured, where: values <1.1 indicate contexts with comprehensive high quality of land; values between 1.1 and 1.2 and between 1.2 and 1.3 are two intermediate levels (the former as medium and the latter as medium to high levels) with decreasing quality of land; and SDI values >1.3 indicate the worst land quality with regard to sensitivity to degradation. For each level of these soil indices, relative values (expressed as a percentage) representing the ratio of the surface area occupied by each zoning class to the total area occupied by all five zoning classes were calculated.

## 3. Results

### 3.1. Soil Quality Index (SQI)

Figure 3 shows different values of associated land zoning classes with the SQI levels for Rome and Barcelona, expressed in terms of both actual surface area (hectares in the table) and relative values (percentage in the bar plot). To enable accurate comparison between Rome and Barcelona, it should be noted that the total amount of natural areas, included in zoning class 5, is almost 20-fold higher in Rome (72,739 ha) than in Barcelona (3898 ha), mainly because Rome’s municipal territory (1285 km^2^) is much larger than that of Barcelona (101 km^2^). Considering each zoning class, conservation-protection land (class 1) occupied a larger land area associated with a high degree of soil quality in Barcelona (169 ha) than in Rome (89 ha). In both absolute and percentage terms, the SQI results revealed that land with the highest soil quality was within the environmental context (class 5) in Rome (~56%), but within the consolidated area (class 2) in Barcelona (~59%), followed by class 5 (~34%). However, based on SQI, around 80% of land with low soil quality was found in the environmental system (class 5) in Rome (Figure 3), while in Barcelona, around 86% of land with low soil quality was found in the consolidated context (class 2) (Figure 3).

### 3.2. Sensitivity to Desertification Index (SDI)

Figure 4 shows the surface area (hectares in the table) and relative values (percentages in the bar plot) of different associated land zoning classes with SDI levels for the case cities. Based on this figure, the largest surface area in Rome was occupied by the consolidated zone (class 2), particularly for land with low SDI (<1.1 as 11,698 ha), but also for land with medium to high SDI (between 1.2 and 1.3 as 12,935 ha). In addition, the environmental zone (class 5) occupied a large surface area in total, and for medium to high SDI (between 1.2 and 1.3 as 54,160 ha) in particular. In percentage terms, the SDI interval between 1.2 and 1.3 (medium to high degradation risk or land sensitivity) was represented significantly by the environmental zone (class 5), while the restructuring and/or requalification zone (class 3) occupied land with low SDI (<1.1 reflecting low degradation risk or land sensitivity). In Barcelona, similar patterns emerged. The largest land area (4818 ha) was associated with the consolidated zone (class 2), followed by the environmental network surrounding the city (class 5) (1962 ha), particularly for land with medium to high and high SDI (>1.2). In percentage terms, all the zoning classes were associated with medium to high and high SDI values (1.2–1.3 and >1.3), indicating moderate-high sensitivity to desertification, although the environmental system (class 5) had a high percentage of surface area falling within the high SDI level (71%).

## 4. Discussion

The present study evaluated the sustainability of land use planning in two Southern European cities, Rome and Barcelona, using classical methodology for land quality comparison and classification in accordance with land management [9,18,68]. Preserving soil quality and mitigating land degradation is a pivotal aspect of local planning and environmental management in both cities [18,63,68]. A holistic evaluation based on two commonly used indices (SQI and SDI) showed that land use planning in Barcelona has taken better account of soil quality context for land allocation into different zoning classes than in Rome. Larger areas of land with low soil quality in Barcelona have been allocated to the consolidated zoning class and larger areas of land with high soil quality have been allocated to environmental systems. Regarding land sensitivity to desertification, there has been large-scale allocation of land with low sensitivity to desertification to restructuring zones with potential for further development in Rome. In both cities, large percentages of land areas with low sensitivity to desertification fall within conservation-protection areas (zoning class 1), while large percentages of land areas with high sensitivity fall within environmental systems areas.

The current structure of both cities includes open spaces that, although highly fragmented, would be suitable for non-intensive agricultural production within a network of agro-forest relict land [3,69,70,71]. Peri-urban areas in both cities are characterized by production potential that is currently underestimated, as it is threatened by the low value of farmland and land dynamics driven by real estate speculation [18]. Rome and Barcelona were found to have different average values of SQI, partly reflecting the approximately 12-fold difference in size of their metropolitan areas (1285 and 101 km^2^, respectively). However, in Rome, natural areas were shown to occupy a 20-fold larger area than in Barcelona. Conservation-protection areas (zoning class 1) occupied a greater area of land with high soil quality in Barcelona (169 ha), but this zoning class still occupied about 89 hectares of high-quality soils in Rome.

More effective integration of economic, societal, and environmental dimensions into land use planning would ensure the achievement of truly sustainable development paths at both regional and local scales [9]. Land use planning critically affects the functionality of cities. Cities in the Mediterranean region, e.g., Hellenic cities, have been developed without or with partial respect to the land use planning framework over past decades. Quantification of land use pattern morphology and spatial configuration can provide a better understanding of urbanization issues, including urban sprawl or compactness [72]. The results in this study also suggest that analysis of urban patterns and processes can benefit from comparative studies of cities characterized by different degrees of planning regulation, age, urban structure, and socioeconomic constraints [71,73]. Spatio-temporal trends in land take indicators can provide a basis for assessing formal (and informal) urban expansion, and support design of policies for urban containment and sustainable growth [64]. One limiting factor for implementation of comparative studies cross-nationally is the low availability of Europe-wide datasets with common standards. Monitoring and assessment activities mainly focus on consequences of urban expansion (e.g., environmental challenges) and not on drivers of urbanization, which highlights the need for more urban datasets with common standards across European cities [71].

Urban expansion into fertile and productive soils would require continuous attention through, e.g., interventions to support the stability of ecosystem balance and sustainable land use [3]. It is necessary to effectively challenge the loss of fertile soils through the development of calibrated medium- and long-term intervention strategies. In this regard, data integration that can be achieved using composite indices, e.g., SQI and SDI, as considered in this study, would support SSP aimed at achieving intelligent soil management and correct urban (and peri-urban) management.

Soil quality and land sensitivity to desertification are issues of current importance which should be included in future scenarios and regional strategies [7,18,63,68]. Since land consumption and degradation processes are the most pronounced threats to urban sustainability, particularly through sealing of high-quality soil [74], knowledge of spatially variable multifaceted relationships between urban expansion, metropolitan structure and land use is critical. Prevention and mitigation actions must be increasingly targeted at areas to be protected [75], both for their soil quality value and for their greater vulnerability [76]. The present comparison of two Mediterranean cities in Europe revealed that they have similar needs, but linked to place-specific factors [77,78]. The intrinsic fragility of Mediterranean landscapes due to soil degradation, recurring drought and aridity, forest fires, and anthropogenic pressures [79,80,81] requires serious reflection on the irreversible transformation of agricultural soils and the loss of land providing local food and ecosystem services [15,69,76]. Land use planning priorities in the Mediterranean region should thus be re-directed towards socio-economic and environmental sustainability, while taking into account the associated implications for local characteristics of Mediterranean cities (e.g., the ability of metropolitan areas to attract investment [82]. Giving a new socio-environmental role to green spaces in metropolitan regions can provide a great opportunity to halt the irreversible transformation of fringe land and the ‘consumption’ of fertile soils.

## 5. Conclusions

Using Rome and Barcelona as examples of intense urbanization and ecological fragility, this study investigated whether recent land use planning in Southern Europe has been oriented towards conservation of soil quality and mitigation of desertification risk. The results obtained highlight the necessity for integration of socio-economic and environmental dimensions into land use planning in the study region, where the fragility of the landscapes raises concerns about irreversible transformation and loss of agricultural land providing food and ecosystem services. This creates a great need to re-orient land use planning priorities towards socio-economic and environmental sustainability. The comparison and analysis performed in this study drew special attention to the need for designing and investigating urban expansion scenarios in land use planning according to environmental characteristics that are sensitive to complex trajectories of development. This can improve SSP for complex socio-economic and environmental systems.

## Figures and Tables

**Figure 1 ijerph-17-04001-f001:**
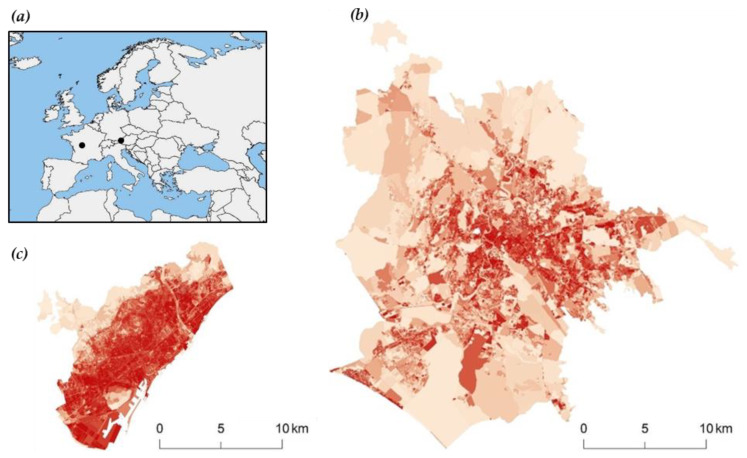
(**a**) Geographical location of the city of Rome in Italy and the city of Barcelona in Spain (black dots). Degree of soil sealing in (**b**) Rome and (**c**) Barcelona. More intense red tone indicates more intense soil sealing rate.

**Figure 2 ijerph-17-04001-f002:**
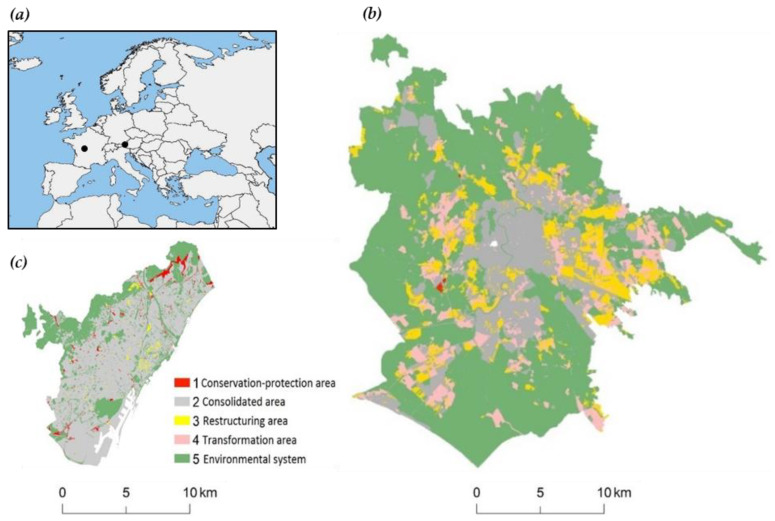
(**a**) Geographical location of the two cities of Rome in Italy and Barcelona in Spain (black dots); Spatial distribution of zoning classes 1–5 (see Table 1) based on the strategic plans for (**b**) Rome and (**c**) Barcelona.

**Figure 3 ijerph-17-04001-f003:**
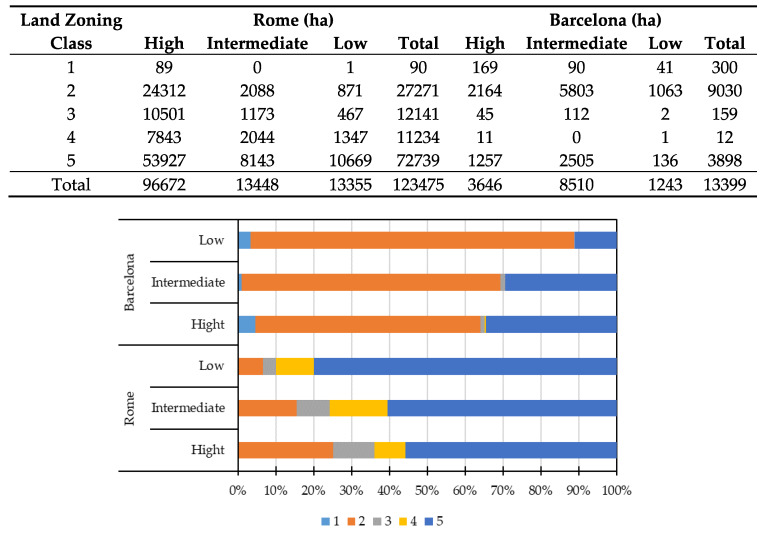
Soil quality index (SQI) values represented by zoning classes 1–5 (see Table 1) in hectares (upper table) and percentages (bar plot) in Rome and Barcelona.

**Figure 4 ijerph-17-04001-f004:**
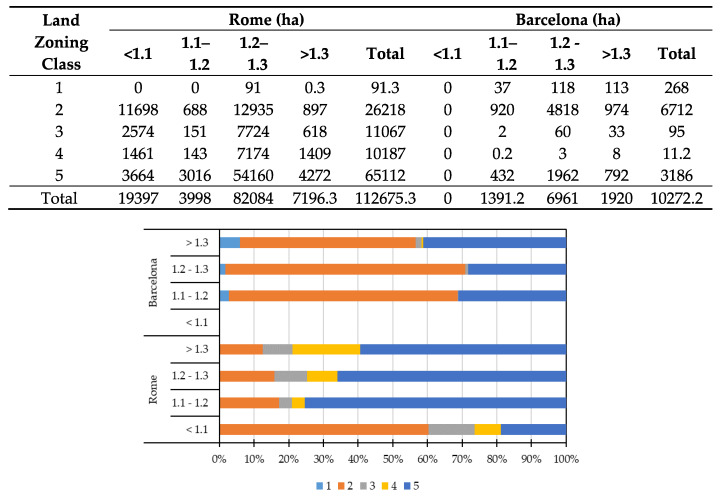
Sensitivity to desertification index (SDI) values represented by zoning classes 1–5 (see Table 1) in hectares (upper table) and percentages (bar plot) in Rome and Barcelona.

**Table 1 ijerph-17-04001-t001:** Land use nomenclature system (zoning classes 1–5) adopted in the present study.

Zoning Class	Code	Description of Land-Use (Zoning) Classes in the Strategic Plan for:	
		Rome	Barcelona
Conservation-Protection Area	1	Areas of Constrained Transformation of the Environmental System, Services, Infrastructure; Structuring Projects	Areas of Urban-Mixed Conservation, not Urbanizable, Protection Systems, Rustic
Consolidated Area	2	Service and Infrastructure System Settlement System—Including Consolidated Area and Historical City	Commercial/Service Settlements; Urban: Including Economic Activity, Services, industrial, Residential-Isolated houses, Residential-Grouped Houses, Residential-Traditional Urban, Residential-Old Town; Systems: Including equipment, Railway, Public Housing, Services, Roads and Urban Soils
Restructuring Area	3	Settlement System: Private/Public Spaces to be Restored/Redeveloped (Urban Infill); Local Central Units	Urban-Mixed
Transforming Area	4	Settlement System: Urban Transformation	Buildable, Including Residential Development; Residential, Open and Mixed Spaces; Systems: Including Ports, Roads, Structural Axes
Environmental System	5	Water, Agro Romano, Protected Natural Areas	Systems, Including Green Areas, Coastal, Hydrographic

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
