# Peer review of "Unraveling Latent Aspects of Urban Expansion: Desertification Risk Reveals More"

_ijerph, 2020, doi:10.3390/ijerph17114001_

Round 1

Reviewer 1 Report

The article presented by Egidi, G. et al. To be published in the International Journal of Environmental Research and Public Health is a good job that, in my opinion, deserves to be published only with a very minor revision.

Authors are correct in describing the state of the art, the methods and the results obtained when evaluating the attention paid in the geographical context of large Mediterranean cities and urban sprawl (focused on Rome and Barcelona as study cases) to aspects such as soil quality or sensitivity to desertification.
Although the methods used are well known and have already been used in other contexts by different authors, generally outside urban settings; this paper succeeds in implementing the knowledge to evaluate aspects of spatial planning in relation to environmental conservation and soil quality in urban areas.
In the discussion section, authors make interesting contributions regarding the need to better integrate the economic, social and environmental dimensions in the design of more respectful plannings, as regards to soil quality.
The English style is good, in my opinion, although I do not consider myself qualified enough to judge this aspect.
Reference section are adequate, covering the main articles, particularly in aspects referred to the development and application of methodologies for evaluating soil / environmental quality and sensitivity to degradation / desertification.

For these reasons, I consider that the manuscript fulfill the requirements of the journal, its editorial line, and deserves to be published without major modifications.

Just. two minor issues:

  • Figure 3 needs to be improved. Y axis don't look properly
  • Reference 56 if Lavado Contador, J. F..........(not J. L.)

Best regards

Author Response

The authors would like to thank you for the thorough consideration and comments. The manuscript has been extensively revised according to all comments of the reviewers and the two minor comments are implemented in the revised version and highlighted with red font color. Below, the reviewer’s comments are listed in black and specific responses to each are provided in blue.

Point 1. Figure 3 needs to be improved. Y axis don't look properly.

Response 1: Figure 3 has been modified and errors on the Y axis have been rectified in the revised manuscript (lines 300-302).

Point 2. Reference 56 if Lavado Contador, J. F..........(not J. L.)

Response 2: The name of the first author in the reference in question (now number [68]) has now been corrected (line 552):

  1. Lavado Contador, J.F.; Schnabel, S.; Gutiérrez, A.G.; Fernández, M.P. Mapping sensitivity to land degradation in Extremadura. SW Spain. Land Degrad. Dev. 2009, 20(2), 129-144.

Reviewer 2 Report

The work that is proposed addresses a subject of interest in a serious and academic way. An important issue linking urban growth and environmental impacts in a specific and fragile urban context: the Mediterranean city through a case study of a Spanish and an Italian example.
However, for publication the article still has to correct many aspects and undertake major revisions. The summary and introduction, a state of the art, are fine.
In Materials and Methods we have a fundamental question. Because the municipality of Barcelona is used as a reference and not its metropolitan area, as most studies do. The municipality was historically amputated for political reasons, and no similar study often uses it. Justify in a much more convincing way because it has been decided to compare a municipality associated only with a compact city with another one spread across a large area of ​​less developed hinterland. What is the relevance of the comparison?
Furthermore, the bibliography used on Barcelona is scarce and not very relevant. A greater effort of scientific knowledge of the city should be made through authors such as H. Capel, O. Nel.lo, J. Busquets, M. Delgado, etc.
The Results are limited to applying the proposed index. But it is a somewhat descriptive section, scarcely analytical or of debate.
Finally, the results seem like conclusions. They are a little repetitive with respect to the text. They are somewhat of a summary and are not too different from the conclusions.
In short, a series of aspects of the text are provided that should be greatly improved. Undertaking these reforms, the article could be considered.

Author Response

The authors would like to thank you for the thorough consideration and useful comments and suggestions. The manuscript has been revised according to all comments. All concerns raised by reviewer #2 are specifically addressed, and all associated changes are highlighted with red font color within the revised version. Below, all comments are listed in black and specific responses to each are provided in blue.

Point 1. In Materials and Methods, we have a fundamental question. Because the municipality of Barcelona is used as a reference and not its metropolitan area, as most studies do. The municipality was historically amputated for political reasons, and no similar study often uses it. Justify in a much more convincing way because it has been decided to compare a municipality associated only with a compact city with another one spread across a large area of less developed hinterland. What is the relevance of the comparison?

Response 1: We wanted to highlight that generally, in comparison studies, it is almost impossible to use and focus on exactly comparable areas from different aspects. Although the cities of Rome and Barcelona are different in size and located in different countries where the concept and definition of urban areas, growth capacities and local socio-economic contexts are distinctive, they are at least located at a similar latitude (geographical similarity) and share a comparable experience of the past urban expansion process (lines 138-139 in the revised manuscript). The main reasons for selecting these two cities for the present comparison were their land use planning system and local government competences in terms of land zoning considerations. Thus instead of selecting areas with similar size, population density, or urban planning system, we selected areas with similar administrative partitions (local municipalities) that have similar competence in land use planning and land zoning classification. Following this criterion, the cities of Rome and Barcelona were selected as being substantially comparable with regard to administrative prerogatives. However, these cities are inherently different regarding the size of their local administration and socio-spatial patterns. The manuscript highlights the importance of integrating socio-economic and environmental dimensions into land use planning to ensure sustainability (as mentioned in lines 347-349 and 383-385). Comparing these two cities in terms of their land use planning, considering soil characteristics, greatly supported the main message of the study. Rome is a large city with generalized discontinuous-dense settlements (as also mentioned by the reviewer), representing large cities in Southern Europe, such as Madrid or Seville. On the other hand, Barcelona is a small city with mostly compact settlements and restricted sprawl hotspots, which is similar to Athens, Lisbon, Salonika or Naples. However, both cities use the same criteria for land use planning. Thus, comparison of the two provided a refined overview of the relationship between land use planning, land zoning, soil quality and desertification risk in vastly different morphological and functional contexts across Southern Europe.

To address this concern of the reviewer, we have included the following explanation in the revised manuscript (lines 179-186):

“While the two cities differ in terms of size, population density, urban expansion capacity, and local socio-economic contexts, their experiences of past urban expansion processes are comparable. They are also both competent with regard to land use planning and land zoning considerations, which makes them substantially comparable regarding their administrative prerogatives. Overall, their comparison in this study provides a refined overview of the relationship between land use planning, land zoning systems, soil quality, and desertification risk in different morphological and functional contexts across Southern Europe.”

Point 2. Furthermore, the bibliography used on Barcelona is scarce and not very relevant. A greater effort of scientific knowledge of the city should be made through authors such as H. Capel, O. Nel.lo, J. Busquets, M. Delgado, etc.

Response 2: Thanks for these suggestions. Three relevant references from these authors, along with some additional literature mainly focusing on and relevant to Barcelona, are included in the revised manuscript, to provide a broader perspective on previous studies and their findings related to the city. However, in terms of most of their great contribution, language was a barrier to accessing their findings. Their focus was more at the local scale and most of their scientific articles are published in Spanish. We experienced the same difficulty in the case of Rome, with many publications in Italian. This makes it a challenge to include local contributions in an English publication with international co-authors. However, we include the most relevant sources in the revised manuscript (lines 168, 197 and 199):

  1. Busquets Grau, J. Barcelona: evolución urbanística de una capital compacta. MAPFRE, Barcelona, 1992.
  2. Capel, H. El debate sobre la construcción de la ciudad y el modelo Barcelona. Scripta Nova. Revista electrónica de geografía y ciencias sociales 2007, 11(229-255).
  3. Nel.lo, O. Coping with metropolitan dynamics. The metropolitan plan of barcelona. Scripta Nova 2011, 15.
  4. Garcia-Ramon, M.D.; Albet, A. Pre-Olympic and post-Olympic Barcelona, a ‘model’for urban regeneration today? Environ. Plan. 2000, 32(8), 1331-1334.
  5. Marshall, T. Regional planning in Catalonia. Europ. Plann. Stud. 1995, 3(1), 25-45.
  6. Marshall, T.C. Environmental planning for the Barcelona region. Land Use Policy 1993a, 10(3), 227-240.

Point 3. The Results are limited to applying the proposed index. But it is a somewhat descriptive section, scarcely analytical or of debate.

Response 3:  The results and discussion sections have been extensively revised in the new version of the manuscript. In this study, we compared the spatial distribution of environmental indicators (focusing on soil) with a shared nomenclature of ‘land zoning’ in land use planning systems for two different cities. In the results section, we present the actual/raw data obtained by applying the proposed methodology in this study. These results consist of a quantitative table with an illustrative bar plot for each of the indices, showing the association of land zoning classes to different levels of the indices. The analysis in the Results section is mainly based on comparison of the two case cities in terms of soil quality and sensitivity to desertification, pointing out the main quantitative results in the tables. These results are considerably discussed from a broader perspective in the following discussion section. In this comparative study, the results were partly descriptive, but in fact they revealed how the two case cities differ in land use planning and associated soil quality considerations.

Point 4. Finally, the results seem like conclusions. They are a little repetitive with respect to the text. They are somewhat of a summary and are not too different from the conclusions.

Response 4: As mentioned in response to the previous point, the results and discussion sections have been extensively revised. The Results section presents the data obtained for the two environmental indices used and associated comparative descriptions for the two case cities. The Discussion section elaborates upon the results from a broader perspective and debates the possible implications for land use planning and environmental policy through highlighting the necessity of integrating socio-economic and environmental aspects into land use planning to support further sustainability. The Conclusions section has been reorganized and revised for more clarity and brevity. There are no specific repetitions in the revised manuscript. Please see the associated changes in the discussion (lines 328-336 and 376-379) and conclusions (lines 389-392) sections.

Reviewer 3 Report

It is an interesting and novel paper. The analysis is quantitative and this makes the discussion of the Conclusions interesting. The Abstract provides a representative summary of the paper and the paper's title is well chosen. The paper “Unraveling Latent Aspects of Urban Growth: Desertification Risk Reveals More” would be appealing for the audience of the International Journal of Environmental Research and Public Health.The paper is well-organized and well written with the appropriate length.

The study discusses the urban growth results in peri-urban socio-spatial systems. It focuses on peri-urban soils degradation, that drives to an increasing desertification risk in ecologically fragile districts. The empirical part of the study concerns the Mediterranean cities of Barcelona and Rome, as emblematic cases of dispersed urbanization and ecological fragility.

There are some minor issues that the authors could consider and possibly address.

[1] Figure 1. Degree of soil sealing in the city of (a) Rome and (b) Barcelona. A more intense red tone indicates a more intense soil sealing rate. I consider that you need to illustrate the 2 Maps in the same scale, in order to avoid misunderstanding of the soil sealing representation.

Probably it is also better to choose only the Built-up area from the soil sealing data set (Sealed cells more than 80%).

[2] Figure 2. Spatial distribution of zoning classes 1-5 (see Table 1) based on the strategic plans for (a) Rome and (b) Barcelona . The same issue, I consider that you need to illustrate the 2 Maps in the same scale in order to avoid misunderstanding concerning the shape and the spatial distribution of zoning.

[3] Apart from the socioeconomic dynamics and the driving forces, the spatial planning framework has important effects on the Urban Growth development mode, which is without question recognized in the present paper. In that context, I consider that this needs to be discussed further, citing some examples in a short paragraph.

[4] Within this framework, I consider that mentioning briefly in the Discussion : (a) Some differences of the European cities character and the Urban Growth difference mode and (b) Referring to examples of other national cases, concerning the urban growth mode and the urban sprawl, could help for a more integrated and international understanding of the unraveling latent aspects of urban growth, and its socio-environmental impact. Possibly this needs to be discussed further in the Conclusions chapter so that the paper’s argument obtains a more international interest.

For example refer to the works of:

1. Stathakis D., Tsilimigkas G., 2015. “Measuring the compactness of European medium-sized cities by spatial metrics based on fused data sets”. International Journal of Image and Data Fusion, Volume 6, Issue 1/2015, Pages: 42-64. (DOI: 10.1080/19479832.2014.941018)

2. Tsilimigkas G., Stathakis D., Pafi M., 2015. “Εvaluating the land use patterns of medium-sized Ηellenic cities. Urban Research and Practice”, Volume 9, Issue 2/2016, Pages: 181-203: 181-203 (DOI:10.1080/17535069.2015.1125940)

3. ChorianopoulosI., Pagonis T., KoukoulasS., Drymoniti S., 2010 “Planning, competitiveness and sprawl in the Mediterranean city: The case of Athens” Cities Volume 27, Issue 4, August 2010, Pages 249-259

Author Response

The authors would like to thank you for the thorough consideration and useful comments and suggestions. The manuscript has been extensively revised according to all comments, all concerns raised by reviewer #3 are specifically addressed, and all associated changes are highlighted with red font color within the revised version. The figures are revised in order to improve readability and quality. Below, all comments are listed in black and specific responses for each are provided in blue.

Point 1. Figure 1. Degree of soil sealing in the city of (a) Rome and (b) Barcelona. A more intense red tone indicates a more intense soil sealing rate. I consider that you need to illustrate the 2 Maps in the same scale, in order to avoid misunderstanding of the soil sealing representation. Probably it is also better to choose only the Built-up area from the soil sealing data set (Sealed cells more than 80%).

Response 1: Figure 1 has been redrawn following the reviewer’s suggestion (lines 203-205). A small insert with a map indicating the geographical location of the studied cities is also included in Figure 1 in the revised manuscript. We used a complete color ramp for soil sealing rate in order to illustrate the entire spatial variability in soil sealing observed in the studied cities. We have not restricted the illustration to only a subsample of soil sealing (e.g. > 80%) in order to be able to highlight and make a comparison for residential settlements and vertical and compact structures in the cities as well. These important parts can include small gardens and occasional green infrastructures, which is especially common in Rome, around the historical center, and would be also relevant to the aims of this study.

Point 2. Figure 2. Spatial distribution of zoning classes 1-5 (see Table 1) based on the strategic plans for (a) Rome and (b) Barcelona. The same issue, I consider that you need to illustrate the 2 Maps in the same scale in order to avoid misunderstanding concerning the shape and the spatial distribution of zoning.

Response 2: Figure 2 has been rescaled following the reviewer’s suggestion (lines 282-284). A small insert with a map indicating the geographical location of the studied cities has been included in both Figures 1 and 2 in the revised manuscript.

Point 3. Apart from the socioeconomic dynamics and the driving forces, the spatial planning framework has important effects on the Urban Growth development mode, which is without question recognized in the present paper. In that context, I consider that this needs to be discussed further, citing some examples in a short paragraph.

Response 3: The interaction between land use planning and urban expansion is highlighted in several parts in the revised manuscript (lines 53-56, 86-88, 109-111, 193-195, 347-349, and 392-397). Also, an additional paragraph has been included in the revised manuscript (lines 79-86) based on the following relevant references as below:

“Land use planning is an appropriate tool for achieving more sustainable use of land and influencing urban expansion over time. For instance, using nature-based solutions (NBSs) in land use planning provides cost-effective long-term solutions for land degradation [36]. Exploring the potential of NBSs and employing them for land-related risk mitigation require in turn improved land use planning and management strategies [37]. In addition, flood risk in urban areas might increase under the impact of land use changes. Conversion of natural areas to impermeable surfaces for urban expansion results in lower infiltration rates and increased surface runoff, which in turn increase the flooding risks in urban regions and threaten urban expansion [38].”

  1. Keesstra, S.; Nunes, J.; Novara, A.; Finger, D.; Avelar, D.; Kalantari, Z.; Cerda, A. The superior effect of nature based solutions in land management for enhancing ecosystem services. Sci. Tot. Environ. 2018, 610-611, 997-1009.
  2. Kalantari, Z.; Ferreira, C.S.S.; Keesstra, S.; Destouni, G. Nature-based solutions for flood-drought risk mitigation in vulnerable urbanizing parts of East-Africa. Curr. Opin. Environ. Sustain. 2018, 5, 73-78.
  3. Kalantari, Z.; Sörensen, J. Link between land use and flood risk assessment in urban areas. Proceedings 2019, 30, 62.

Point 4. Within this framework, I consider that mentioning briefly in the Discussion: (a) Some differences of the European cities character and the Urban Growth difference mode; and (b) Referring to examples of other national cases, concerning the urban growth mode and the urban sprawl, could help for a more integrated and international understanding of the unraveling latent aspects of urban growth, and its socio-environmental impact. Possibly this needs to be discussed further in the Conclusions chapter so that the paper’s argument obtains a more international interest. For example, refer to the works of:

  1. Stathakis D., Tsilimigkas G., 2015. “Measuring the compactness of European medium-sized cities by spatial metrics based on fused data sets”. International Journal of Image and Data Fusion, Volume 6, Issue 1/2015, Pages: 42-64. (DOI: 10.1080/19479832.2014.941018)
  2. Tsilimigkas G., Stathakis D., Pafi M., 2015. “Εvaluating the land use patterns of medium-sized Ηellenic cities. Urban Research and Practice”, Volume 9, Issue 2/2016, Pages: 181-203: 181-203 (DOI:10.1080/17535069.2015.1125940)
  3. ChorianopoulosI., Pagonis T., KoukoulasS., Drymoniti S., 2010 “Planning, competitiveness and sprawl in the Mediterranean city: The case of Athens” Cities Volume 27, Issue 4, August 2010, Pages 249-259

Response 4: The suggested references have been included in the revised manuscript. Also, according to their key findings the following further explanations have been included in the Discussion (lines 349-353, 357-362 and 380-383) as suggested by the reviewer:

“Land use planning critically affects the functionality of cities. Cities in the Mediterranean region, e.g., Hellenic cities, have been developed without or with partial respect to the land use planning framework over past decades. Quantification of land use pattern morphology and spatial configuration can provide a better understanding of urbanization issues, including urban sprawl or compactness [73].”

“One limiting factor for implementation of comparative studies cross-nationally is the low availability of Europe-wide datasets with common standards. Monitoring and assessment activities mainly focus on consequences of urban expansion (e.g., environmental challenges) and not on drivers of urbanization, which highlights the need for more urban datasets with common standards across European cities [76].”

“Land use planning priorities in the Mediterranean region should thus be re-directed towards socio-economic and environmental sustainability, while taking into account the associated implications for local characteristics of Mediterranean cities (e.g., the ability of metropolitan areas to attract investment [85].”

  1. Tsilimigkas, G.; Stathakis, G.; Pafi, M. Measuring the compactness of European medium-sized cities by spatial metrics based on fused data sets. Urban Research & Practice 2016, 9(2), 181-203.
  2. Stathakis, D.; Tsilimigkas, G. Measuring the compactness of European medium-sized cities by spatial metrics based on fused datasets. International Journal of Image and Data Fusion (IJIDF) 2015, 6(1), 42-64,
  3. Chorianopoulos, I.; Pagonis, T.; Koukoulas, S.; Drymoniti, S. Planning, competitiveness and sprawl in the Mediterranean city: The case of Athens. Cities 2010, 27, 249-259.

Reviewer 4 Report

The article addresses an interesting topic related with the role of land use planning and strategic planning in the regulation of urban expansion and its specific impact on what authors called “soil quality and mitigation of desertification risks”.  However, there are a number of conceptual and methodological aspects that need to be clarified in order to strengthen the argumentative and explanatory structure of the research (sections 4 and 5):

  1. It is necessary to distinguish among spatial planning (set of methods and approaches), land use planning (process to regulate the land to promote specific outcomes) and strategic planning (process to define intervention priorities). L81, L.84, L.93, L.176. If the idea is to incorporate land use and strategic planning tools in the analysis, the current "development vision" (L.100) (result of strategic planning) for Barcelona is unclear (section 2.1).
  1. Urban growth, urban expansion, and urban sprawl are used synonymously throughout the text even though they differ conceptually. Please clarify.
  2. Revise the wording used in the research objective: “emblematic cases” (L.23), “recent spatial planning” and “land zoning revisions” (L.24-25), explaining their meaning and relevance.
  3. What is a planning support system and how it is related with land use planning? (L.41)
  4. What is meant by “practical planning”? (L.88)
  5. The paragraph (L.110-117) is very confusing. Please clarify.
  6. Which is the “recent SSP orientation” (L.121). Land use plans and strategic plans were launched at the same time for the same period in both cities?
  7. “Cities were selected because they are surrounded by traditional rural landscapes experiencing increasing ecological fragility and…”(L.121-123). Please add scientific evidence. Moreover what means “traditional” rural landscapes?
  8. The city of Rome is or was unique for its unspoilt peripheral countryside (L. 138)? See further argumentation (L.143 to 149). Please clarify.
  9. It is important to analyse, the goals, area and the criteria by which land planning official documents were made (section 2.2.1). These antecedents might prevent research bias and may support conclusions.
  10. Land nomenclature system developed to standardized zoning classes needs more reflection and clarity. For instance, land conservation and land protection are different concepts (class 5 includes protected natural areas). Class 3 restructuring refers to urban infill? What implies and includes a “non- urbanised land mainly used for commercial activities” (L.237).

Author Response

The authors would like to thank you for the thorough consideration and useful comments and suggestions. The manuscript has been extensively revised according to all comments, all concerns raised by reviewer #4 are specifically addressed, and all associated changes are highlighted with red font color within the revised version. Below, all comments are listed in black and specific responses to each are provided in blue.

Point 1. It is necessary to distinguish among spatial planning (set of methods and approaches), land use planning (process to regulate the land to promote specific outcomes) and strategic planning (process to define intervention priorities). L81, L84, L93, L176. If the idea is to incorporate land use and strategic planning tools in the analysis, the current "development vision" (L100) (result of strategic planning) for Barcelona is unclear (section 2.1).

Response 1: Based on this comment, the whole manuscript has been revised in terms of terminology. The unique term “land use planning” is used, as it was related to our research context. However, the strategic spatial planning (SSP) concept is addressed in the Introduction, not as the focus in our research, but to indicate the consolidated delay in Southern European contexts in applying these kind of instruments/visions compared with other European contexts. The SSP concept is mentioned in the Discussion and Conclusions sections to demonstrate more clearly the potential significant contribution of our analytical approach to definition of intervention priorities for urban sustainability. As such, the sentence including “vision development” has been changed to “Strategic spatial planning (SSP) is mainly intended as an integrated and more sustainable development approach…” (lines 105-106). Our study was not aimed at incorporating land use and strategic planning tools, but rather highlighting the necessity for integrating environmental indicators into land use and strategic spatial planning.

Point 2. Urban growth, urban expansion, and urban sprawl are used synonymously throughout the text even though they differ conceptually. Please clarify.

Response 2: The whole manuscript has been revised in terms of terminology for further clarity and coherence, using uniquely the noun “urban expansion” as it was the main focus of our study. The title of the manuscript has also been changed accordingly, from “Unraveling latent aspects of urban growth…” to “Unraveling latent aspects of urban expansion…”. In some cases, when necessary, we have qualified the term “urban expansion”, e.g. by adding appropriate adjectives such as ‘dispersed’ (lines 21 and 152) or ‘deregulated’ (line 116,).

Point 3. Revise the wording used in the research objective: “emblematic cases” (L23), “recent spatial planning” and “land zoning revisions” (L24-25), explaining their meaning and relevance.

Response 3: In the revised manuscript, the term “emblematic cases” has been changed to “examples” (line 22). The terms “recent spatial planning” and “land zoning revisions” have been revised to “land use planning” (line 23) as a more general term for both. The whole manuscript has been rechecked and all such terms revised.

Point 4. What is a planning support system and how it is related with land use planning? (L41)

Response 4: The term “planning support system” has been changed to “decision support system” in the revised manuscript (line 91).

Point 5. What is meant by “practical planning”? (L88)

Response 5: The term “practical planning” has been changed to “land use planning” in the revised manuscript (line 94).

Point 6. The paragraph (L110-117) is very confusing. Please clarify.

Response 6: This paragraph has been revised for greater clarity (lines 112-122).

Point 7. Which is the “recent SSP orientation” (L121). Land use plans and strategic plans were launched at the same time for the same period in both cities?

Response 7: The term “recent SSP orientation” has been removed in the revised manuscript since it was not really focused upon and assessed throughout our study. We have avoided entering into too detailed an assessment of local planning technical rules, which was outside the scope of our study. We use simple methodology to clearly highlight the implications of land use planning and environmental status in Mediterranean region with regard to soil quality. The novelty of the work lies in the integration of basic land use planning (land zoning) and environmental indicators for the Mediterranean region (with informal urban development up to the 1980s and less influence of participatory planning practices on final land decisions). This manuscript provides a new interpretation of the relationship between (settlement) form and (environmental) functions.

Point 8. “Cities were selected because they are surrounded by traditional rural landscapes experiencing increasing ecological fragility and…”(L121-123). Please add scientific evidence. Moreover, what means “traditional” rural landscapes?

Response 8: The sentence has been extended in the revised manuscript to include some characteristics of the traditional rural landscape, along with a scientific reference (lines 126-130):

“These cities were selected because they are surrounded by traditional rural landscapes (the Mediterranean agro-forest mosaic typical of lowland/coastal districts and mixing extensive tree crops (olives and vineyards), arable and garden crops, and relict woodland), experiencing increasing ecological fragility and land sensitivity to degradation under climate change and human pressures [50].”

  1. Biasi, R.; Brunori, E.; Smiraglia, D.; Salvati, L. Linking traditional tree-crop landscapes and agro-biodiversity in Central Italy using a database of typical and traditional products: A multiple risk assessment through a data mining analysis. Biodivers. Conserv. 2015, 24(12), 3009-3031.

Point 9. The city of Rome is or was unique for its unspoilt peripheral countryside (L. 138)? See further argumentation (L143 to 149). Please clarify.

Response 9: The associated sentence has been removed in the revised manuscript for clarity, brevity and to improve coherence. A new supporting statement has been included (lines 144-148):

“The municipal territory is heterogeneous, with mixed impervious and semi-natural land contrasting with the compactness of the historical center, where the most important functions are concentrated. However, with residential mobility and suburbanization, Rome’s morphology has been progressively transformed toward more polycentric and spatially balanced settlements.”

Point 10. It is important to analyze, the goals, area and the criteria by which land planning official documents were made (section 2.2.1). These antecedents might prevent research bias and may support conclusions.

Response 10: As mentioned for the previous points, we avoided entering into a detailed assessment of local planning technical rules and their intended goals, which was outside the scope of our study. However, some relevant references are included in section 2.2.1 to address point 10. The additional references discuss extensively the land use planning and land zoning technique in the case cities. Due to some language barriers (the official documents are mainly in Italian for the case of Rome and in Spanish for the case of Barcelona), we have not included further explanations about land use planning official documents in the revised manuscript. Instead, we refer to the most relevant available documents for detailed information:

  1. Blanco, I.; Bonet, J.; Walliser, A. Urban governance and regeneration policies in historic city centres: Madrid and Barcelona. Urban Research & Practice 2011, 4(3), 326-343.
  2. Garcia-Ramon, M.D.; Albet, A. Pre-Olympic and post-Olympic Barcelona, a ‘model’for urban regeneration today? Environ. Plan. 2000, 32(8), 1331-1334.
  3. Marshall, T. Regional planning in Catalonia. Europ. Plann. Stud. 1995, 3(1), 25-45.
  4. Marshall, T.C. Environmental planning for the Barcelona region. Land Use Policy 1993a, 10(3), 227-240.

Point 11. Land nomenclature system developed to standardized zoning classes needs more reflection and clarity. For instance, land conservation and land protection are different concepts (class 5 includes protected natural areas). Class 3 restructuring refers to urban infill? What implies and includes a “non- urbanised land mainly used for commercial activities” (L237).

Response 11: Table 1, including the land use nomenclature system, has been completely revised (line 280). The terms in the table have been rechecked and revised according to the land use planning official documents (although the local language barrier still applied). The table reports the original land classification for both case cities as stated in their plans. In some cases, it was difficult to exactly translate the land zoning terminology from Italian and Spanish/Catalan into correct and truly understandable English terms. However, the land zoning systems were rather similar for the two case cities. We have tried to present the associated information in Table 1 in an illustrative and coherent structure for land zoning classes to the official documents (linguistically). This would allow technical reproducibility of the same information in other similar contexts (e.g. in other cities belonging to the ‘Latin’ European world, such as Portugal, Southern France and, in some cases, even Greece). In addition, the information presented in Table 1 is intended to be fully understandable to international audiences from ‘non-Latin countries’. We have added a new reference [51], which discusses extensively the land zoning system specifically for Rome (line 248). Relevant references are also included in the revised manuscript for Barcelona. With these revisions, we have further explained the criteria we adopted in the study.

  1. Clemente, M.; Zambon, I.; Konaxis, I.; Salvati, L. Urban growth, economic structures and demographic dynamics: exploring the spatial mismatch between planned and actual land-use in a Mediterranean city. Int. Plann. Stud. 2018, 23(4), 376-390.

Round 2

Reviewer 2 Report

The authors have responded to the majority of comments and recommendations made to them in the first review.
Therefore, the publication of the work is now recommended.

Reviewer 4 Report

XXXXXXXXXXXXXXXXXXXXXXXXXXXthe authors have adequately incorporated the authors have adequately incorporated the comments madethe comme

The authors have adequately incorporated the comments made.